# Pain education for patients with non-specific low back pain in Nepal: protocol of a feasibility randomised clinical trial (PEN-LBP Trial)

Saurab Sharma,[1,2] Mark P Jensen,[3] G Lorimer Moseley,[4] J Haxby Abbott[2]

¹Department of Physiotherapy, Kathmandu University School of Medical Sciences, Dhulikhel, Bagmati, Nepal
²Centre for Musculoskeletal Outcomes Research, Dunedin School of Medicine, University of Otago, Dunedin, Otago, New Zealand
³Department of Rehabilitation Medicine, University of Washington, Seattle, Washington, USA
⁴School of Health Sciences, University of South Australia, Adelaide, South Australia, Australia

**Correspondence to**
Saurab Sharma;
saurabsharma1@gmail.com

## ABSTRACT

**Introduction** Low back pain (LBP) is the leading cause of years lived with disability in Nepal and elsewhere. Management of LBP that is evidence-based, easily accessible, cost-effective and culturally appropriate is desirable. The primary aim of this feasibility study is to determine if it is feasible to conduct a full randomised clinical trial evaluating the effectiveness of pain education as an intervention for individuals with LBP in Nepal, relative to guideline-based physiotherapy treatment. The findings of the study will inform the planning of a full clinical trial and if any modifications are required to the protocol before undertaking a full trial.

**Methods/analysis** This protocol describes an assessor-blinded feasibility clinical trial investigating feasibility of the pain education intervention in patients with non-specific LBP in a physiotherapy hospital in Kathmandu, Nepal. Forty patients with LBP will be randomly allocated to either pain education or guideline-based physiotherapy treatment (control). Outcomes will be assessed at baseline and at a 1 week post-treatment. The primary outcomes are related to feasibility, including: (1) participant willingness to participate in a randomised clinical trial, (2) feasibility of assessor blinding, (3) eligibility and recruitment rates, (4) acceptability of screening procedures and random allocation, (5) possible contamination between the groups, (6) intervention credibility, (7) intervention adherence, (8) treatment satisfaction and (9) difficulty in understanding the interventions being provided.

**Ethics/dissemination** The protocol was approved by Nepal Health Research Council (NHRC; registration number: 422/2017) and University of Otago Human Ethics Committee for Health (registration number: H17/157). The results of the study will be presented at national and international conferences and published in a peer-reviewed journal.

**Trial registration number** NCT03387228; Pre-results.

## INTRODUCTION

Low back pain (LBP) is a highly prevalent health condition worldwide.[1 2] It is the leading cause of disability[2] and imposes huge economic burden to the society in both developed and developing countries.[3–6] LBP is among the most common health conditions contributing to years lived with disability in Nepal.[2] Although the prevalence of LBP is high in Nepal, ethnographic research has noted that LBP-related disability may be low in rural areas,[7] perhaps due to the very low socioeconomic status of individuals living in rural Nepal, which forces them to keep working despite the presence of pain. Consistent with this idea, another study highlighted that 80% of people with chronic pain in Nepal continue to work.[6] However, it is alarming that number of spine surgeries for spinal pain has been increasing in Nepal over the years,[8] despite lack of evidence supporting efficacy for this treatment.[1 9]

### Interventions for management of LBP

Many interventions have been investigated for the management of LBP. These include surgery,[10 11] pharmacotherapy,[12–17] exercises,[18–20] advice for self-management including advice to remain physically active[11] and psychological therapies.[21 22] As alluded to earlier, biomedically focused interventions such as surgery and pharmacotherapy are not recommended for a non-specific LBP as the evidence does not support their effectiveness.[1 14] Moreover, they are associated with significant risks for adverse events and are costly.[5]

Clinical practice guidelines for LBP recommend self-management including

reassurance, education and advice to remain active as the first line of care that should be provided to all the patients with LBP. Superficial heat and manual therapy (massage/manipulative therapy) are recommended for acute LBP, whereas exercise and psychological therapies are recommended for chronic LBP.[23–25]

**Strengths and limitations of this study**

► Assessor and statistician will be blinded to group allocation of the participants.
► Therapist and patient blinding is not possible in this study because of the nature of the treatments provided.
► Conclusions regarding the effectiveness of the intervention cannot be made because of the feasibility design of the current study.

## Pain education for LBP

Patient education for LBP that has been investigated in randomised controlled trials is basically of two types: biomedical education and pain biology education.[26] The first refers to educating patients about vertebral anatomy and pathoanatomy of the spine, which has been shown to be ineffective and may even have negative effects on LBP outcomes.[26] However, the second type of education—pain biology education (hereafter called as 'pain education')—has been shown to have positive effects on both pain and disability.[27 28] Pain education is structured education programme with specific aims and objectives.[29] This intervention has a list of target key concepts to be delivered and includes the curriculum contents to deliver the key concepts using up-to-date pain science knowledge, stories and metaphors.

It has been previously hypothesised that this type of education programme using metaphors and stories may be an effective intervention in Nepalese with chronic pain.[30] However, the pain education materials that have been developed in western cultures are not necessarily valid and equally effective in reducing pain and disability in non-western cultures. Therefore, when developing pain education materials in a newer language or culture, (significant) cultural adaptations of the education materials may be required to make it suitable for the target population, as culturally inappropriate education may not produce desirable results.

Therefore, in order to evaluate the effectiveness of pain education in individuals with non-specific LBP from Nepal, culturally appropriate pain education materials should first be developed for Nepal, specifically. However, it is possible that the adaptations made could potentially reduce its effectiveness. Thus, before testing the adapted pain education in a full clinical trial, a feasibility study is needed in order to determine if a full clinical trial based on the adapted intervention is warranted, or if additional modifications may be needed prior to performing the full trial.

## Why the feasibility trial?

We propose a feasibility trial because: (1) the intervention (ie, pain education) will need significant cultural adaptation, although it has been evaluated for efficacy previously in other languages and western cultures; (2) the adapted intervention has never been investigated for its efficacy or effectiveness before; (3) the population in question (individuals with extremely low socioeconomic status and educational attainment in Nepal) is unique; and (4) a high-quality clinical trial in individuals with LBP has not been conducted in Nepal to our knowledge, and

we therefore do not know if a full trial is feasible. The findings from the proposed feasibility study will inform the planning and design of a full trial, if the results indicate that a full trial is warranted.

The results of the full trial will have significant clinical implications for the management of LBP in Nepal and similar cultures, providing empirical evidence if pain education is a viable treatment for the management of LBP, and if it is effective in reducing pain, disability and emotional distress.

## Aims and objectives

The primary aim of the study is to evaluate the feasibility of a full randomised clinical trial (RCT) for assessing the effects of pain education as an intervention for patients with LBP of any duration in a physiotherapy facility in Nepal after developing culturally appropriate, evidence-based pain education materials. The primary objectives of the study are related to feasibility of an RCT, specifically: (1) willingness to participate in an RCT, (2) feasibility of assessor blinding, (3) eligibility and recruitment rates, (4) acceptability of screening procedures and random allocation, (5) possible contamination between the groups, (6) intervention credibility for patients with LBP, (7) intervention adherence, (8) treatment satisfaction and (9) difficulty in understanding the intervention being provided.

## METHODS AND ANALYSIS
### Study design and setting

This is a feasibility study that is being performed to determine if a full RCT can be successfully conducted using the procedures and protocol of the feasibility study, or if modifications of the protocol are needed prior to conducting the full trial. The study findings will inform the design of the full trial, if the trial is found to be feasible.[31]

The definition of a feasibility study highlights the question, '*Can this study be done?*'. The Standard Protocol Items: Recommendations for Interventional Trials (SPIRIT) statement,[32] the Consolidated Standards of Reporting Trials statement extension to pilot and feasibility randomised trial[31] were followed in the planning of the study and reporting of the protocol.

The study will be an assessor-blinded, two-arm, feasibility RCT. The study is registered in Clinicaltrials.gov (trial registration number: NCT03387228). The study will be conducted in the Sahara Physiotherapy Hospital, Kathmandu, Nepal.

## Overview of the study

Advertisement of the trial will be made in social media, and all the patients with LBP presenting at the study site will be invited to participate. Interested candidates will be screened for eligibility. Eligible patients with non-specific LBP will then be enrolled in the trial and be randomly assigned to one of the two study groups. All the participants in the experimental group will receive pain

education and those in the control group (CG) will receive guideline-based physiotherapy treatment. All the participants will be assessed at baseline and 1 week following treatment. Details describing the schedule of enrolment, interventions and assessment are presented in table 1, in the manner recommended by SPIRIT checklist.[32]

**Table 1** Schedule of enrolment, assessment and interventions

| Timepoint | Enrolment $-T_1$ | Allocation $T_0$ | Post allocation $T_1$ | $T_2$ | Final Assessment $T_3$ |
|---|---|---|---|---|---|
| **Enrolment** | | | | | |
| Eligibility screening | X | | | | |
| Explain study procedure/provide participant information sheets | X | | | | |
| Informed consent | X | | | | |
| Random treatment allocation | | X | | | |
| Intervention | | | | | |
| Experimental intervention (PEG) | | | | X | |
| Control group (CG) | | | | X | |
| Assessment | | | | | |
| Baseline descriptive variables | | | | | |
| Sociodemographic information (age, sex, address, occupation, religion and ethnicity) | | | X | | |
| History | | | | | |
| Pain history (site of pain, duration of pain, continuous or intermittent pain, aggravating and relieving factors) | | | X | | |
| Comorbidities | | | X | | |
| Feasibility | | | | | |
| Willingness to participate in a randomised controlled trial | X | | | | |
| Acceptability of random allocation to one of the two groups | X | | | | |
| Acceptability of intervention session (one session in a week with home treatment programme throughout the week) | X | | | | |
| Feasibility of blinding the assessor* | | | | | X |
| Eligibility and recruitment rates | X | | | | |
| Acceptability of screening procedures* | | | | | W |
| Understanding possible contamination between the groups | | | | | X |
| Evaluating the credibility of the intervention | | | X | X | |
| Adherence to intervention | | | | | X |
| Treatment satisfaction | | | | | X |
| Difficulty in understanding the treatments | | | | | X |
| Secondary outcomes | | | | | |
| PROMIS Pain interference | | | X | | X |
| PROMIS Pain intensity | | | X | | X |
| Quality of life | | | X | | X |
| PROMIS Sleep disturbance | | | X | | X |
| PROMIS Depression | | | X | | X |
| GROC | | | − | | X |
| PCS | | | X | | X |
| CD-RISC-10 | | | X | | X |

*Assessed by the therapist providing intervention; all other outcomes are assessed by the blinded outcome assessor.
CD-RISC-10, 10-item Connor-Davidson Resilience Scale; GROC, Global Rating of Change; PCS, Pain Catastrophizing Scale; PROMIS , Patient-Reported Outcome Measurement Information System; $T_1$, enrolment time; $T_0$, allocation time; $T_1$ , baseline assessment (before treatment); $T_2$, during treatment; $T_3$, 1 week post-treatment; W, assessment at the end of every week on Fridays.

 

## Participants

Patients with non-specific LBP seeking rehabilitative services at Sahara Physiotherapy Hospital will be invited to participate. Interested patients will be screened by a research assistant (physiotherapist by training) involved in the research.

### Inclusion criteria

Non-specific LBP (LBP other than those excluded, see exclusion criteria below) of any duration with pain primarily localised between T12 and gluteal folds, in patients aged 18 years or older, with average pain intensity reported as moderate, severe, or very severe on a Patient-Reported Outcome Measurement Information System (PROMIS) five-point PROMIS Pain Intensity Short-form Scale[33] over the past week, and who is a Nepalese and is able to understand and speak Nepali fluently will be included in the study.

### Exclusion criteria

Participants with likely specific causes of LBP will be excluded using a triage procedure as suggested by Bardin and colleagues.[34] This includes exclusion of participants having history of prolonged use of corticosteroid, history of malignancy, recent history of fever or chills, history of other diseases associated with compromise in immune system, history of recent spinal surgery or dental procedures, recent history of trauma to spine or a fracture of a spine, history of bladder and bowel dysfunction, history of perineal or saddle anaesthesia and history of weakness of lower extremity or loss of sensation in lower extremity. Additionally, current pregnancy and history of diagnosed mental health conditions that would limit adherence to the trial procedures will be excluded.

### Sample size

For a feasibility study, it is inappropriate to calculate sample size based on desired statistical power to detect a treatment effect,[35] because the primary aim of the study is to assess if a full trial can or should be conducted. Feasibility outcomes are descriptive in nature; therefore, inferential statistics regarding treatment effects will not be computed. To achieve the primary objectives related to feasibility outcomes, the research team estimated that 40 participants would be adequate.[36] Twenty patients will be randomly allocated to each treatment condition.

### Participant screening and recruitment

Consecutive participants with non-specific LBP will be invited to participate in this study. The study purpose and procedures will be described to potential participants. This will include information about the benefits and potential harms of the intervention, the time required for the completion of the study, follow-up duration, voluntariness of participation, cost of participation and the rights to withdraw from the study at any point. A study information sheet will be provided to all potential participants.

If the potential participants are interested in participating, they will be screened for eligibility by a research assistant who is a physiotherapist. If the participants are found eligible, informed consent will be obtained. For those who cannot sign the consent, a witness will sign on their behalf, or the study participant will provide a thumb print on the form for those who cannot write or sign the form as per the ethical guidelines provided by Nepal Health Research Council (NHRC). We will include uneducated patients who cannot sign an informed consent in order to increase the inclusion of uneducated or low education group, given that 31% people in Nepal who are 5 years old or more cannot read and write.[37] Additionally, exploration of feasibility of pain education in those with no schooling or low educational attainment is important in order to inform clinical practice.

Participants will be informed that they will receive one of the two treatments randomly. It will be highlighted that both of the treatment options are thought to be effective for LBP and that the goal of the main study is to compare the interventions; however, the current study will more specifically evaluate the feasibility of such a study.

### Group allocation, randomisation and blinding

Random number sequence, in random blocks of 4 and 6, will be generated using www.randomization.com, by a researcher (JHA) who is not involved in recruitment process. Allocation concealment will be performed using opaque, sealed envelopes. The participants will be allocated to one of the two groups by a hospital staff member who is not the assessor. The two groups will be: pain education group (PEG) and CG.

### Intervention

The Template for Intervention Description and Replication Checklist was followed when planning the study intervention.[38 39] Manuals of standard operating procedures will be followed during the delivery of the interventions in both the groups. This will ensure treatment uniformity and fidelity. It is not possible to blind the intervention providers based on the design of the study.

Participants in intervention group (PEG) will receive detailed pain education as described in the next paragraph, and those in the CG will receive guideline-based physiotherapy treatment. After the completion of the post-treatment assessment at 1 week, study participants in both the groups will receive the treatment being provided by physiotherapists at Sahara Physiotherapy Hospital. Participants in both the groups will be encouraged not to seek for other medical care for LBP during the 1-week study period, unless they have to. If they do undertake other forms of treatment, they will be requested to report this during the follow-up assessment, and this will be recorded.

### Pain education group

The pain education will be delivered to the PEG only. We will use the pain education handouts in Nepali for Nepalese with LBP based on the resources developed by Moseley and Butler, called Explain Pain.[29 40 41] It

has evolved in over 15 years[42] and undergone changes and advances.[29 40 41 43 44] Pain education is delivered to provide reassurance, which means removal of fear and concerns about illness.[45] Reassurance is among the core aspects of management of patients with non-specific problems such as LBP.[46] Although there are no gold standard ways to provide reassurance and alleviate fear and concerns about a disease or its consequences, empathy and collaboration are thought to play important role.[47] The Pain education intervention will be provided by the lead investigator (SS) who is trained in the delivery of this treatment and has about 10 years of experience in the management of musculoskeletal disorders, including LBP.

In order to develop the pain education resources in Nepali, the first step is the development of curriculum for pain education for patients with LBP[29] in Nepal. The curriculum was first outlined in English (by SS) in the similar manner recommended by the developers[29] and was reviewed and approved by both the developers (see online supplementary appendix 1). Based on the curriculum, the pain education materials and patient handouts were created (by SS) in Nepali. The clinical cases and pain stories that were compiled are actual stories collected from patients and clinicians in Nepal but will be anonymously shared to provide reasoning based on contemporary pain biology knowledge.[29] Pictures that are found in Explain Pain resource materials[29 40 41] were adapted in the Nepali version.

The pain education handout and materials that are produced in Nepali is proofread and will first be pretested in 5–10 Nepalese with LBP as needed, and corrected, if necessary, before using in the feasibility study participants. The final handouts will then be printed for the use in the current study. This adapted process will ensure that the Nepali pain education materials produced are valid and culturally appropriate. However, for the purpose of the full trial, the difficulty in understanding the treatment will be assessed in the current feasibility study, and any modifications required will be made.

### Dosage of the PEG intervention

A single approximately 1 hour pain education will be delivered to the PEG, because evidence indicates that: (1) interventions as brief as 5 min have been shown to have reassuring effects lasting for up to a year,[44] and a single consultation has been found to be as reassuring as the multiple session interventions.[47] We prefer a single session treatment over multiple session treatment, because for many patients in rural Nepal, it is difficult to deliver a multiple session treatment in reality. Thus, we plan to provide a single session delivering key concepts of Explain Pain, which will be reinforced by providing patient handouts to look at and read at home. In fact, a 1-hour session should be adequate to cover the key Explain Pain concepts by keeping the content simple and jargon free. Use of a plain language in the health-related education is important to adapt in low health literacy.[48]

### Home advice for PEG group

A printed handout of the pain education material will be provided only to the study participants in the PEG. Participants will be advised to read them, and perform physical activity including walking for approximately 30 min. Education accompanied by written information has been reported to yield the largest effects on fear component of emotional distress.[47] However, education level of Nepalese is low (65%);[37] we will adapt the written materials to incorporate many more images than text.

For those who cannot read, the family member(s) will be encouraged to read out the materials to them. To complement the written materials, we will also provide an audio-recording of the pain education session to the patients as an online URL link to Nepali patient education material stored in the cloud or will be copied to their smart phones or both for those who have the facilities to use them. The pain education advice will be directed towards reducing brain's perception of movement and exercise as a threat encouraging participants to slowly pace the movement, physical activity and exercise. This is thought to desensitise the sensitive nervous system and improve function. A written instruction to perform general exercises and physical activity will be sent to the participants. Participants will be discouraged to use a lumbar corset and rest as coping strategy, whereas physical activity and return to work will be encouraged. A reminder to perform home exercises will be sent to all participants for a total of 5 days in the week.

### Control group

The intervention will be provided by the physiotherapists working at the study site. Treatment integrity in the CG will be improved by providing an interactive seminar to all the physiotherapists delivering CG intervention by the lead researcher. The seminar will incorporate evidence-based information, including assessment and management of LBP based on the current recommendations from clinical practice guidelines.[23 49–51] Research articles and simplified evidence-based summary will be provided to the physiotherapists before the interactive seminar for self-study. At the end of the seminar, a brief multiple-choice quiz will be conducted for the study physiotherapists that will assess evidence-based management of LBP. The therapists will need to score a minimum of 80% before they deliver treatment to the CG.

### Intervention in the CG

The control participants will receive physiotherapy care based on the recent clinical practice guidelines from three different western countries: (1) American College of Physicians, USA (2017),[23] (2) National Institute for Health and Care Excellence, UK (2016)[49 51] and (3) Toward Optimized Practice, Canada (2015).[50] These guidelines are used because they are recent and highly regarded. We did not find any evidence-based clinical practice guidelines for the management of LBP in Nepal or other developing countries. The CG interventions were derived by

comparing the recommendations made by each of the three guidelines on specific management strategies. The CG treatment components were selected if: (1) two or more guidelines recommended the treatment component, (2) all the authors of the current study agreed that the component should be included, (3) the component was culturally acceptable and feasible to deliver in Nepal, (4) the component was determined to be appropriate to deliver at first contact and (5) the total duration of all of the components would sum up to 1 hour (approximately) to make the contact hour comparable with the approximately 1 hour of pain education that would be provided to the experimental group. Similarities and differences in the clinical practice guidelines across the countries are common and are influenced highly from the experience of the expert committee responsible for developing a clinical practice guideline and local practice trends.[52] For this reason, we chose to adapt the recommendations made in the available guidelines to fit with the expertise of the local physiotherapist, practice trends that are widely accepted in Nepal and the feasibility of delivering the treatment within the context of the study.

Thus, the CG intervention will contain of: (1) education (advice to remain active, education about prognosis of LBP for acute LBP and avoid bed rest and braces)[23 50 51] for 10–15 min, including time spent to listen to each participant's pain story; (2) back massage[23 51] for about 10 min; (3) superficial heat[23 50] for 10–15 min; and (4) static cycling or (treadmill) walking with the aim to promote physical activity[23 50 51] including any rest period for a total of approximately 20 min. Although superficial heat is recommended in the acute/subacute LBP,[23 50] we included this as a common treatment for all types of LBP including chronic LBP, which could be a part of self-management. Any forms of electrotherapy and acupuncture will not be offered to the study participants.[49] The control intervention will strictly exclude the use of pain biology education.

### Dosage of intervention for the CG
The control intervention will last for 1 hour to match the experimental group.

### Home advice for the CG
Participants will be advised to self-manage their back pain based on the information provided. Home exercise leaflet with emphasis on the value of exercise to increase strength and endurance, followed by a 30 min walking. Advice preceding the exercises will state that exercises are needed to keep you strong, healthy and pain free. A written instruction to perform general exercises and physical activity will be sent to the participants. Participants will be discouraged to use a lumbar corset and rest as coping strategy, and return to work and physical activity will be encouraged in the control participants, as they are in the experimental group. A reminder to perform home exercises will be sent to all the participants for 5 days during the week.

### Outcome measures
The details of primary feasibility outcomes are presented in table 2.

All but one secondary outcome measures have been shown to be reliable and valid in Nepali populations. The measure of quality of life (QOL) is not yet validated at the time of writing the protocol; however, we have included it as a secondary outcome measure, because assessment of QOL is a recommended measure by core outcome sets in clinical trials for LBP.[53] We hypothesise that this measure is comprehensible and will show adequate validity in the Nepalese sample. We will evaluate the validity of the measure before using it in the full clinical trial.

A research assistant will interview all the study participants to make the study procedures consistent and to allow for the inclusion of participants with little or no education. The interviews will be administered by a physiotherapist who will be blind to group assignment. Secondary measures include the four PROMIS short-form measures assessing pain interference,[54] pain intensity,[54] sleep disturbance,[54] and depression,[54] as well as the 13-item Pain Catastrophizing Scale,[55] Connor-Davidson Resilience Scale,[56] Global Rating of Change[57 58] and a QOL scale. All the items in each PROMIS measure will be summed to obtain raw scores for each scale. The raw scores of each measure will then be converted to T-scores, with a mean of 50 and SD of 10 and recorded (www.assessmentcenter.net). Details of the measures with their measurement properties are presented in table 3.

The risk of adverse events in both the groups are very low. Participants will be asked to choose the amount of home exercises (such as walking) they will perform based on a level that is comfortable. Participants will be asked to change the duration and/or pace of exercises if they feel the initial level is too high.[59] Participants will be asked to record any adverse events that occur and report these to the researcher. Adverse events in both the groups will be reported and compared between the groups.

### Additional measures
Additional questionnaires will be administered to obtain data related to: (1) sociodemographic information (age, sex, education level, employment status, income, religion and ethnicity); (2) pain history, including duration of pain, aggravating and relieving factors, other associated comorbidities; and (3) pain location using pain drawings. Other information such as resources required to conduct the trial (eg, cost) and time required to complete the recruitment of desired number of participants will also be recorded.[60] Total duration of home exercises in each group will also be recorded.

### Criteria for feasibility
The results of this feasibility trial will indicate if the study as designed is feasible, which will inform the decision of progressing to a full trial with the recommendations. The decision will be one of the following: (1) do not proceed

**Table 2** How will each primary feasibility objective be assessed?

| Serial Number | Objectives | Measures to assess specific objectives | Statistical analysis |
|---|---|---|---|
| 1 | Willingness to participate in a randomised controlled trial | All the consecutive participants presenting at the physiotherapy department will be approached and invited to participate. The number and rate of participants willing versus not willing to participate will be recorded, as well the reasons for not wanting to participate. | Total number of patients not willing to participate will be summed and reasons for non-participation collated. |
| 2 | Feasibility of blinding the assessor | Feasibility of blinding will be assessed by asking the assessor two questions at the end of the follow-up assessment:<br>1. Did you receive any information that indicated to you which group the participant was assigned to?<br>2. How did you receive information about group assignment?<br>Assessor's guess regarding group assignment group will be recorded for each participant. The responses will be coded as correct or incorrect guess. | The frequency and relative rates of 'Yes' and 'No' as the answer to the first question will be computed, as will the frequency and relative rates of correct and incorrect guesses. Finally, reasons for guesses will be recorded as verbatim and reported. |
| 3 | Eligibility and recruitment rates | The total number of participants screened, found eligible and recruited will be recorded during the screening and recruitment process. The reasons why the participants were ineligible for study inclusion will be recorded for every participant who did not meet eligibility criteria. Reasons for declining participation will be recorded for every eligible individual who declined participation. Consent rates will also be recorded. | The number of participants screened, who were eligible, who consented to participate, who refused to participate will be computed. The reasons for exclusion and refusal will also be collated. |
| 4 | Acceptability of screening procedures | The lead researcher will interview the physiotherapist(s) screening the potential participants for eligibility by asking the following open-ended question at the end of every week, 'Were there any difficulties or challenges in screening and recruiting the participants the past week?' The responses will be recorded as 'Yes' or 'No'. If the answer to this question is 'Yes', the lead researcher will ask the following open-ended question, 'What were the difficulties and challenges?'. Responses will be written down verbatim. Additionally, the outcome assessor will complete a form every time he or she encounters a difficulty or a challenge during screening. Outcome assessors will also be asked about their recommendations for overcoming any challenges that they identify.<br>Time taken to complete the questionnaires will be recorded. | The frequency of difficulties or challenges will be counted. The difficulties and challenges will be noted, categorised if possible, and reported. Results will be used to improve the screening procedure in the full trial in future, and therapists' recommendations will be considered. |
| 5 | Acceptability of random allocation to a treatment group | Outcome assessor will ask the participants if random allocation to one of the two treatment groups is acceptable to the participants. Responses will be recorded as 'Acceptable', 'Not acceptable' or 'No preference'. | The frequencies of each response will be computed separately for each treatment condition. |
| 6 | Understanding possible contamination between the groups | Contamination between the groups will be assessed by asking all the study participants the following questions at 1 week following treatment:<br>1. 'Have you talked to other participants in this study about the intervention they are receiving?'; If yes, further ask, 'Was your attitude towards the intervention, or intervention changed after talking to one of the participants in the other group?';<br>2. 'Are you aware of the intervention that any participants in the other group are receiving in the study?';<br>3. Are any of the participants in the other group aware of the type of intervention you are receiving in this study?'<br>Participants in the control group will be asked the following question: 'Did you read the pain education booklet or watch the video which is one of the components of the intervention group?'<br>These questions are adapted from a cotwin controlled feasibility trial.[62] | The frequencies of the participants who responded affirmatively to each question will be computed separately for each treatment condition. |

Continued

**Table 2** Continued

| Serial Number | Objectives | Measures to assess specific objectives | Statistical analysis |
|---|---|---|---|
| 7 | Credibility and acceptability of the interventions | Treatment credibility will be assessed using five questions adapted from Borkovec and Nau.[63] The questions will be modified to fit 'Pain education' and 'Control group' as a treatment for patients with LBP and will be administered to all the participants in both conditions. One, important cultural adaptation for the scoring will be made by changing the numerical scale proposed by Borkovec and Nau to a Verbal Rating Scale, because Numerical Rating Scales have been shown to be unsuitable in Nepalese with musculoskeletal pain, especially in those who are older and have low educational attainment.[64] Responses for each of the five questions will be recorded on a Likert scale where, 0='Not at all', 1='A little bit', 2='Somewhat', 3='Quite a bit', 4='Very much'. The questions are: 1. How logical does the treatment provided seem to you for the management of low back pain? 2. How confident would you be that this treatment would be successful in reducing pain? 3. How confident would you be in recommending this treatment to a friend or family who also has LBP? 4. If you were having low back pain again, would you be willing to undergo such treatment? 5. How successful do you feel this treatment would be in reducing pain in other parts of the body, for example, knee pain? All the questions will be asked at baseline and at 1 week following treatment. | Mean of the total scores on the credibility scale will be computed separately for each condition. Between-condition differences in creditability will be evaluated using a t-test. |
| 8 | Adherence to the intervention | Adherence to treatment will be assessed by asking the participants in both conditions to maintain a record of daily home treatment that was followed by every participant. The participants will tick mark a box for every day for a total of 5 days between the two assessment time-points to indicate their adherence to prescribed home exercise programme. Additionally, participants will be requested to record any deviation of prescribed home treatment programme (eg, additional visit to the clinic and use of other interventions). Finally, question regarding non-adherence will be asked. | The total number of treatment adherence days will be summed for each treatment condition separately. Deviation from the treatment protocol will be recorded. Finally, the reasons for non-adherence will also be listed. |
| 9 | Satisfaction of treatment | All the participants in both the conditions will be asked to respond to the Patient Global Assessment of Treatment Satisfaction scale at 1 week following treatment.[65 66] The question asked will be, 'How satisfied are you overall with the study treatment?' Responses to this question are made on a five-point categorical scale (0='Very dissatisfied'; 1='Dissatisfied'; 2='Neutral or no preference'; 3= 'Satisfied'; 4='Very satisfied'). | Mean treatment satisfaction scores will be computed for each treatment condition separately. Between-condition differences will be evaluated using a t-test. |
| 10 | Difficulty in understanding the information provided by the physiotherapist. | All the participants will be asked about the difficulty in understanding the information provided by the physiotherapist. The question asked will be, 'How difficult was it for you to understand the information provided by the physiotherapist?' Responses will be provided on a 5-point Likert scale, where 1='Very easy', 2='Easy', 3='Neither easy nor difficult', 4='Difficult', 5='Very difficult'. | The differences in the difficulty in understanding the information provided will be compared between the two groups. |
| 11 | Difficulty in understanding the instructions for what to do. | All participants will be asked about the difficulty in understanding the instructions provided by the physiotherapist for what to do at home. The question asked will be, 'How difficult was it for you to understand the instructions provided by the physiotherapist for what to do at home?' using a on a 5-point Likert scale, where 1='Very easy', 2='Easy', 3='Neither easy nor difficult', 4='Difficult', 5='Very difficult'. | Frequencies and rates of each response category will be computed for each group separately. |
| 12 | Adverse events | All participants will be asked about any adverse events after treatment. All responses will be recorded verbatim. | The number of adverse events listed will be computed for each treatment condition separately. The responses will be analysed qualitatively (see text). |

LBP, low back pain.

**Table 3** Secondary outcome measures

| Domain | Name of outcome measure | No. of Items | Response scale | Scoring | Measurement properties of Nepali versions of the scale |
|---|---|---|---|---|---|
| Pain interference | PROMIS Pain Interference short-form 6b[54] | 6 | 5-point, ordinal | Respondents are asked to rate how much pain interfered with their daily life (eg, enjoyment of life, ability to concentrate and day-to-day activities) in past 7 days on 5-point Likert scales (=‘Not at all’, 2=‘A little bit’, 3=‘Somewhat’, 4=‘Quite a bit’, 5=‘Very much’). Responses are scored as a T-score that can range from 0 to 100, with a mean of 50 and SD of 10 in the normative sample. | ▶ Good internal consistency (Cronbach's alpha=0.85). ▶ Excellent reliability (ICC=0.80) in a sample of individuals with chronic pain from Nepal.[54] |
| Pain catastrophising | Pain Catastrophizing Scale (PCS)[55] | 13 | 5-point, ordinal | Respondents are asked to indicate the degree or frequency with which they have each catastrophising response listed when they ‘… are experiencing pain’ on a 5-point Likert scale (0=‘Not at all’, 1=‘To a slight degree’, 2=‘To a moderate degree’, 3=‘To a great degree’ and 4=‘All the time’). The total PCS score can range from 0 to 52, with higher scores representing higher pain catastrophising. | ▶ Good to excellent internal consistencies: Cronbach's alpha=0.85–0.93. ▶ Excellent test–retest stability (ICC=0.89–0.90). ▶ Positive moderate correlations with measures of pain intensity, depression and anxiety in a sample of individuals with chronic pain from Nepal.[55] |
| Patient's global rating of change | Global Rating of Change[57 66 67] | 1 | 1–7, ordinal | Respondents are asked to rate their ‘overall improvement’ their health condition since the study enrolment. Responses can range from 1 to 7 with 4=‘No change’. Scores greater than 4 indicate greater improvement and scores lower than 4 indicate a perceived worsening in the respondent's health condition. | MIC: 1 point change.[57 58] |
| Quality of life (QOL) | QOL rating scale | 2 | 5-point, ordinal | Respondents are asked to rate their general QOL and general health by responding to the questions: ●‘In general, how would you rate your overall quality of life during the past week?’ ●‘How would you rate your general health during the past week?’ The response options are: 1=‘Very bad’, 2=‘Bad’, 3=‘Fair’, 4=‘Good’ and 5=‘Very good’.[68] Greater score indicate better QOL. | Not available during the time of protocol writing the study protocol. |
| Pain intensity | PROMIS Pain Intensity short-form 3b[54] | 3 | 5-point, ordinal | Respondents are asked to rate their current pain and their worst and average pain intensity in past 7 days on 5-point Likert scales (1=‘Had no Pain’, 2=‘Mild’, 3=‘Moderate’, 4=‘Severe’, and 5=‘Very severe’). Responses are scored as a T-score that can range from 0 to 100, with a mean of 50 and SD of 10 in the normative sample. | Good test–retest reliability (ICC=0.71) in a sample of individuals with chronic pain from Nepal.[54] |
| Sleep disturbance | PROMIS Sleep disturbance short-form 8b[54] | 7* | 5-point, ordinal | Respondents are asked to rate sleep quality items on a 5-point Likert scale. Responses are scored as a T-score that can range from 0 to 100, with a mean of 50 and SD of 10 in the normative sample. | Although the Nepali translation of the 8-item Sleep Disturbance short-form showed poor reliability, removal of one item, improved the test–retest stability to excellent. (ICC=0.78). Good internal consistency of 7-items (Cronbach's alpha=0.89).[54] |
| Depression | PROMIS Emotional Distress-Depression short-form 8b[54] | 8 | 5-point, ordinal | Respondents are asked to indicate the frequency of depressive symptoms in the past 7 days on a 5-point verbal rating scale (1=‘Never’, 2=‘Rarely’, 3=‘Sometimes’, 4=‘Often’ and 5=‘Always’). Responses are scored as a T-score that can range from 0 to 100, with a mean of 50 and SD of 10 in the normative sample. | ▶ Excellent internal consistency (Cronbach's alpha=0.93). ▶ Excellent test–retest reliability (ICC=0.81) in a sample of individuals with chronic pain from Nepal.[54] |

Continued

to a full trial if any preplanned changes may not help improve the feasibility; (2) modify the protocol further prior to conducting a full trial; (3) continue with the full trial using the same procedures used in the feasibility trial without modifications; however, monitor the study procedures closely; and (4) continue without modifications, as it is in the feasibility trial, close monitoring is not required.[61] The criteria for the feasibility are presented in table 4.

### Limitation of the study

As this is a feasibility study, the results of the current study will not provide findings regarding the efficacy of the interventions being tested. The study will only evaluate if this research study is viable as a full trial and inform any recommendations for modification of the protocols for the full trial resulting from the findings of this feasibility trial.

### Plan for supervision and monitoring

The study will be conducted and monitored by the lead investigator under the supervision of the coauthors (JHA and MPJ), with assistance from research assistant(s). All the ethical principles as provided by Declaration of Helsinki will be followed by all the members of this research throughout the study. The investigators will not violate any of the rules and ethical principles of NHRC. Monitoring for the NHRC ethical principles will be regulated by the primary investigator and followed by all researchers and research assistants involved in the study.

### Plan for data integrity and management

The research data will be collected by a research assistant who will be trained to collect the research data and manage the data by compiling in a file for individual patient. Participant identifiers (including name, address and contact information) will be removed from the research data and will be stored separately. Data will be entered in Microsoft Excel. Identification of the groups as intervention and CG will be removed from the excel sheet. Research data will be monitored weekly by scrutinising entered data. Any errors in entry will be identified (if any) and amended. Consent forms will be scanned and stored in password-protected computers of the lead researcher and at the University of Otago along with other research data files.

### Data analysis plan

Descriptive statistics will be computed to describe the baseline and demographic characteristics of the study participants. As it is a feasibility study, level of significance and hypothesis testing regarding treatment efficacy will not be performed. Effect sizes representing between-group differences in change in the primary and secondary outcomes will be computed, but these effect sizes will not be considered as a criteria for sample size estimation for the full trial, nor as a criteria to proceed to the full trial, because of the inadequate power of the current feasibility study. Treatment effects for the secondary outcome

**Table 3** Continued

| Domain | Name of outcome measure | No. of Items | Response scale | Scoring | Measurement properties of Nepali versions of the scale |
|---|---|---|---|---|---|
| Resilience | 10-item Connor Davidson Resilience Scale[56] | 10 | 4-point, ordinal | Respondents are asked to rate each resilience item on a 5-point Likert scale: 0='Not true at all', 1='Rarely true', 2='Sometimes true', 3='Often true', and 4='True nearly all the time'. Responses are summed such that total scores range from 0 to 40, with higher scores indicating more resilience. | ▲ Good to excellent internal consistency (Cronbach's alpha=0.87–0.90).[56] ▲ Excellent test–retest stability (ICC=0.89).[56] ▲ SE of measurement=2.42 points. ▲ Minimum detectable change=6.72 points. ▲ Significant negative and moderate association with the PCS in a sample of individuals with chronic pain from Nepal.[56] |
| Use of pain medications and other pain treatments | – | – | – | Names and dosage of pain medication intake will be recorded by the assessor by asking the research participant via interview. Medications will then be categorised into analgesic type (opioids, non-steroidal anti-inflammatory drugs, sedatives and antiseizure medications), as will the number of days each type of medication is taken. Other pain treatments provided to or used by the study participants will also be recorded and classified (eg, physical therapy and naturopathy). The number of days each treatment was provided to or used by the participants will be recorded. | ▲ No validity data for self-reported analgesic or pain treatment use in Nepali patients available at the time of protocol writing. |

*Only seven items out of eight items will be included, as the measurement properties of the total eight-item was poor, and only after removal of one item, the reliability improved.
ICC, Intraclass Correlation Coefficient; MIC, minimum important change; SE, standard error.

**Table 4** Criteria for feasibility

| Criteria | Full trial is not feasible as designed | | Proceed to a full trial without modification in the protocol of the feasibility trial | |
| --- | --- | --- | --- | --- |
| | Modify the protocol prior to a full trial if… | Action | Monitor the study procedures closely if… | Close monitoring is not required if… |
| Blinding of assessor | Assessor has a >70% correct guess rate on the group allocation. | Identify ways to improve assessor blinding based on the responses or feedback provided by the assessors. | 70%–90% blinding is found. | <10% incorrect guess to the group allocation. |
| Recruitment rate | ≤1 participant every week | Identify reasons for low participation rates or declining participation. Possible strategies could be changing the study site, increasing the number of study sites, increasing advertising efforts and using incentives for participation. | 2–3 participants recruited every week. | ≥4 participants recruited every week. |
| Attrition rate (in both arms)[62] | >30% total drop-outs at 1 week post-treatment. | Identify possible reasons for drop-outs and ways to improve follow-up participation. | 15%–30% total drop-outs at 1 week post-treatment. | <15% total drop-outs at 1 week post-treatment. (www.pedro.org.au) |
| Feasibility of outcome assessment | >20% missing data on the secondary outcome measures. | Reduce the number of outcome measures. Identify and use brief versions of outcomes. | 10%–20% missing data on the secondary outcome measures. | <10% missing data on the secondary outcome measures. |
| Contamination of intervention[62] | ≥15% contamination between the groups. | Identify reasons for contamination and resolve them. | <15% contamination between the groups. | 0% contamination between the groups. |
| Score on the credibility of treatment scale | Control condition is >0.50 SD credible than pain education group. | Develop a new control condition for the EP trial and pilot test to ensure that its credibility rating is <0.50 SD different from pain education group. | - | The credibility scores of the two conditions are within 0.50 SD units of each other. |
| Adherence to treatment | <50% attend the treatment session after randomisation. | Identify reasons for not attending the treatment session in order to increase attendance in the full trial. | 50%–80% of the participants attend the treatment session. | 80% or more of the participants attend the treatment sessions. |
| Difficulty scale | If ≥50% participants in the experimental group rate pain education as very difficult, very easy, easy or neither easy nor difficult. | Make the intervention a little more challenging if more than half of the participants report 1–3 out of 5 in the difficulty scale, by increasing the depth of the education, by adding more pain biology education content. Whereas if more than half participants report the intervention as very difficult, reduce the complexity. Re-evaluate complexity in a second cohort of at least five participants to ensure appropriate level of difficulty. | - | ≥50% participants in the experimental group rate pain education as a 'difficult' intervention (4 out of 5 in the difficulty scale). This is the preferred difficulty level because we want pain education intervention to be difficult enough to challenge participants but not too easy or very difficult. |

measures will be presented as means, SD and CIs of the means. Difference between the mean scores of each secondary outcome will be compared with the minimum important change (MIC) values of the outcome measures, if the MIC scores are available. The analysis plans of the primary feasibility objectives are described in table 2.

## Patient and public involvement

The research question was informed by the clinical observation that many patients from rural Nepal showed significant improvement in their LBP outcomes after reassurance and advice to remain physically active. We therefore designed this trial to assess the feasibility of conducting a study to compare the effectiveness of pain education and structured guideline-based physiotherapy treatment in Nepalese people with non-specific LBP.

Patients were not involved in the design of the study protocol but were directly involved in the development of Nepali versions of outcome measures. Patients will also provide feedback and comments in the Nepali pain education materials during pretesting before using it in the feasibility study. Similarly, the development of Nepali Pain Education materials have incorporated real pain-related stories of Nepalese living with pain. The name and identity of all patients were kept confidential. Any information that discloses identity of the patients were excluded in the written pain education booklet.

During the initial assessment, all participants will be asked if they would like to know about the results of the study. A plain language summary of the study results will be written both in English and Nepali, which will be published online. The principal investigator of the study (SS) will also post an audio summary of the research results online for those who cannot read. The link of these will be sent to the participants as text messages.

## ETHICS AND DISSEMINATION

The results of the study will be presented at national and international conferences and published in a peer-reviewed journal.

**Twitter** link_physio

**Acknowledgements** The authors would like to thank Associate Professor David Butler and Tim Cocks from Neuro-Orthopedic Institute (NOI) in Adelaide for their input during the development of curriculum for explaining pain for this study. Dr Adrian Traeger (University of Sydney) provided valuable suggestions in the earlier version of the protocol. Finally, authors would also like to thank all the patients who agreed to use their pain stories in this Pain Education programme.

**Contributors** SS, JHA and MPJ conceived the idea of the trial. Trial was designed by all authors. SS prepared the first draft of the protocol. Successive drafts were edited by JHA, MPJ and GLM. The final version of the protocol was approved by all the authors.

**Funding** The authors have not declared a specific grant for this research from any funding agency in the public, commercial or not-for-profit sectors.

**Competing interests** SS is supported by University of Otago Doctoral Research Grant. GLM has received support from Pfizer, AIA Australia, Gallagher Bassett, Kaiser Permanente USA, Port Adelaide Football Club, Arsenal Football Club and the International Olympic Committee. GLM receives royalties for books on pain and rehabilitation, including the text on which the content for the proposed intervention

was based. He also receives speaker fees for lectures on pain and rehabilitation. Other authors report no competing interests.

**Patient consent** Obtained.

**Ethics approval** The protocol was approved by Nepal Health Research Council (ref number 422/2017) and University of Otago Human Research Committee (Health) (ref number H17/157).

**Provenance and peer review** Not commissioned; externally peer reviewed.

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
