## [Reviewer comments · BMJ Open]

ARTICLE DETAILS

TITLE (PROVISIONAL)	Pain Education for patients with non-specific low back pain in Nepal: Protocol of a feasibility randomized clinical trial (PEN-LBP Trial).
AUTHORS	Sharma, Saurab; Jensen, Mark; Moseley, G.; Abbott, J. Haxby

VERSION 1 – REVIEW

REVIEWER	Kory Zimney University of South Dakota, United States of America
REVIEW RETURNED	08-Mar-2018

GENERAL COMMENTS	Main impression: This is a study protocol for feasibility of RCT for LBP patients in Nepal comparing pain education intervention (experimental) to guideline-based usual care (control). Pain education has shown to be beneficial in other studies compared to usual care. The use of pain education has not been explored specifically in this country and potential cultural and socioeconomic barriers may decrease the effectiveness of the intervention. This protocol for a feasibility study demonstrates an interesting idea that adds to the current knowledge base on this topic. Title: clear and concise Abstract: Concise and specific to topic, with good readability Introduction: Previous pertinent literature cited, discussed, and builds toward purpose of study. Edits: Page 6/line 13: Delete “as long as it is based on accurate neurophysiological knowledge.” I do not think it is established that inaccurate education has negative side effects, and statement does not add to making compelling statement for purpose of study. Page 6/line 26: The first refers to educating patients about vertebral anatomy “and pathoanatomy of the spine”. Methods: Study design is appropriate to achieve study objectives with a few minor revision suggestions to consider strongly. Study population is adequately described, along with allocation and randomization process. Some concern that the one-week follow-up period may not be adequate to see changes in status on many of the outcome measurements. A two or four week period might be more adequate to see changes in some of the outcomes measurements, especially for participants with more chronic/persistent pain problems. It is unclear from the manuscript if the same physiotherapist will be delivering the PEG and the CG, or if different physiotherapist will be providing the care for the PEG compared to the CG. If different physiotherapists are used then therapeutic alliance between participants and physiotherapists could be a significant confounder not accounted for in the study design. Therapeutic alliance has been shown to be a predictor and variable in outcomes with individuals with low back pain.
--

REVIEWER	Balkrishna Bhattarai BP Koirala Institute of Health Sciences (BPKIHS), Dharan, Nepal
REVIEW RETURNED	28-Mar-2018

GENERAL COMMENTS	The protocol manuscript is well written and is about a very important topic. The statistical tests planned for comparison of outcomes at places seem inappropriate and needs to be corrected.
---

REVIEWER	Matheus Almeida Universidade Cidade de São Paulo, Brazil
REVIEW RETURNED	02-Apr-2018

GENERAL COMMENTS	The manuscript aims to evaluate the feasibility of a full randomized clinical trial for assessing the effects of pain education for patients with low back pain in Nepal. Overall, the proposal is interesting; elucidating the importance and how conduct a feasibility study, and it is original in the context of being conducted in low-income country with a non-western culture. The methods section is well structured and reported with sufficient details. However, I would like to raise some concerns: 'Introduction Section'  - I believe the Introduction is a little bit long. I suggest you to remove some information, for example when you talk about the management of LBP (you could resume this information). Sometime, you repeat the same information in the same paragraph (i.e. 'When delivering, to a non-western culture, pain education that was developed in western culture'). I also suggest to remove the last sentence from the section 'Why the feasibility trial'. - Please use the reference from the last GPD, when you talk that LBP is the leading cause of LBP - In the paragraph about clinical practice guidelines, the recommendations reported are for chronic LBP. For acute LBP, the recommendations from the last guidelines are quite different. Please make clear this in the text and use the references from the last guidelines. - You use the reference #26 when say that education is cost-effective intervention. However, this paper did not evaluate the cost-effectiveness of education. Actually, this content has not been totally established in the literature. Maybe it would be more prudent to say that education is likely to be a cost-effective therapy (https://www.ncbi.nlm.nih.gov/pubmed/27550240). - I think you need to make it clear in the introduction and objective section that your focus is patients with LBP with any duration of symptoms. 'Methods Section'  - Please remove the last sentence of the section 'study design and setting'. This information is repetitive (be careful with repetition in all the text). - I don't think you need to say what a feasibility study is. - Why did you decide to include only participants with average pain intensity reported as moderate, severe, or very severe? - Why did you choose to use the PROMIS five-point PROMIS Pain Intensity Short-form Scale? - In the description of the intervention in the education group, I believe you don't need to explain that education is supported by research and the available evidence in this section. So you should remove this paragraph. - I am not sure that you should call the intervention proposed as 'evidence based usual physiotherapy care'. Recent guidelines
---

	emphasize the importance of exercises. Why did you not consider the inclusion of exercise therapy in your treatment group? Once you call it as evidence based treatment, exercise has to be included. The same for manual therapy.
--	--

VERSION 1 – AUTHOR RESPONSE

Reviewer(s)' Comments to Author:

Reviewer: 1

Reviewer Name: Kory Zimney

Title: clear and concise

Abstract: Concise and specific to topic, with good readability

Introduction: Previous pertinent literature cited, discussed, and builds toward purpose of study.

Edits: Page 6/line 13: Delete “as long as it is based on accurate neurophysiological knowledge.” I do not think it is established that inaccurate education has negative side effects, and statement does not add to making compelling statement for purpose of study.

Response: We have removed the text based on reviewer’s suggestion. Thank you.

Page 6/line 26: The first refers to educating patients about vertebral anatomy “and pathoanatomy of the spine”.

Response: Thank you for your suggested edit. We have added the suggested text in the revised manuscript (see page 6, lines 42-43).

Methods: Study design is appropriate to achieve study objectives with a few minor revision suggestions to consider strongly. Study population is adequately described, along with allocation and randomization process. Some concern that the one-week follow-up period may not be adequate to see changes in status on many of the outcome measurements. A two or four week period might be more adequate to see changes in some of the outcomes measurements, especially for participants with more chronic/persistent pain problems.

Response: Thank you for your valuable suggestion. We agree with the reviewer that, for patient outcomes, the follow-up duration of one week may not be enough to detect changes in the outcome measures, and longer follow-up durations would be more appropriate for a definitive study to evaluate the effects of the treatment. However, the objectives of the current feasibility study do not focus on the clinical efficacy but on the feasibility of performing the trial. The shorter duration of follow-up allows us to achieve these feasibility objectives. In the event that the study proves feasible, we will certainly use a longer follow-up period in the full clinical trial.

It is unclear from the manuscript if the same physiotherapist will be delivering the PEG and the CG, or if different physiotherapist will be providing the care for the PEG compared to the CG. If different physiotherapists are used then therapeutic alliance between participants and physiotherapists could be a significant confounder not accounted for in the study design. Therapeutic alliance has been shown to be a predictor and variable in outcomes with individuals with low back pain.

Response: We agree that the issue of therapeutic alliance as a non-specific factor that can impact outcome is important. In the feasibility trial, pain education will be provided by a trained physiotherapist who developed the Nepali Explain Pain materials; the control group treatment will be provided by other physiotherapists working at the center.

The reason we chose to provide the treatment in the two different groups by different physiotherapists was to avoid risk of contamination of the treatment between the groups, if both the group treatments are provided by the same therapist. The advantage of using physiotherapists at Sahara Physiotherapy Hospital to deliver the control group treatment is that it increases the generalizability of results. To retain these benefits, we had to trade with therapeutic alliance. To minimize the effect of

therapeutic alliance or attention bias, the therapists delivering the treatment to both groups will all be Nepalese registered physiotherapists and native Nepali speakers of a similar age, and the total duration of treatment time will be equivalent, at one hour in both the groups. Although therapeutic alliance will not be measured in this feasibility study as this would not address the aims of the study (i.e., it concerns feasibility and not outcome), we do plan on measuring therapeutic alliance in the full trial, in the event that the results of the feasibility trial are positive (i.e., if we determine that a full trial is feasible).

Reviewer: 2

Reviewer Name: Balkrishna Bhattarai

The protocol manuscript is well written and is about a very important topic. The statistical tests planned for comparison of outcomes at places seem inappropriate and needs to be corrected.

Response: Thank you for your review of the protocol manuscript, and highlighting the importance of the topic. It was not entirely clear to us which of our proposed analyses were deemed appropriate and which ones were deemed inappropriate. All of the proposed analyses have been reviewed and recommended by our study statisticians, and we believe would be appropriate for determining feasibility. While we do understand that the proposed analyses are not appropriate for evaluating the efficacy of the interventions, the goal of the current study is to evaluate feasibility, not efficacy.

Reviewer: 3

Reviewer Name: Matheus Almeida

The manuscript aims to evaluate the feasibility of a full randomized clinical trial for assessing the effects of pain education for patients with low back pain in Nepal. Overall, the proposal is interesting; elucidating the importance and how conduct a feasibility study, and it is original in the context of being conducted in low-income country with a non-western culture. The methods section is well structured and reported with sufficient details.

However, I would like to raise some concerns:

'Introduction Section'

- I believe the Introduction is a little bit long. I suggest you to remove some information, for example when you talk about the management of LBP (you could resume this information). Sometime, you repeat the same information in the same paragraph (i.e. 'When delivering, to a non-western culture, pain education that was developed in western culture').

Response: Thank you for your comments. We have revised the manuscript to shorten the Introduction section as per reviewer's suggestion.

I also suggest to remove the last sentence from the section 'Why the feasibility trial'.

Response: We removed the sentence from the revision.

- Please use the reference from the last GPD, when you talk that LBP is the leading cause of LBP.

Response: Thank you for your suggestion. We made the suggested change in the manuscript (see page 4, lines 2-6).

- In the paragraph about clinical practice guidelines, the recommendations reported are for chronic LBP. For acute LBP, the recommendations from the last guidelines are quite different. Please make clear this in the text and use the references from the last guidelines.

Response: Thank you for your comment and suggestions. We have revised the manuscript to clarify the recommendations listed (see page 6, lines 26-31), and updated the citations.

- You use the reference #26 when say that education is cost-effective intervention. However, this

paper did not evaluate the cost-effectiveness of education. Actually, this content has not been totally established in the literature. Maybe it would be more prudent to say that education is likely to be a cost-effective therapy (<https://www.ncbi.nlm.nih.gov/pubmed/27550240>).

Response: Thank you for the suggestion, and noticing the error in the citation. In the revised manuscript, we have removed the text describing cost effectiveness of education in order to reduce the length of the introduction as per previous comment.

- I think you need to make it clear in the introduction and objective section that your focus is patients with LBP with any duration of symptoms.

Response: We added the text in the Introduction section (see page 9, line 102) to clarify that patients with any duration of low back pain were included in the current study.

'Methods Section'

- Please remove the last sentence of the section 'study design and setting'. This information is repetitive (be careful with repetition in all the text).

Response: Thank you for this suggestion. We have removed the suggested text from the revised manuscript.

- I don't think you need to say what a feasibility study is.

Response: We agree with the reviewer that it may not be necessary to say what feasibility study is. However, many readers misunderstand the distinctions between a feasibility study and a full clinical trial, and incorrectly expect the feasibility study to focus on evaluating efficacy. We feel it is worthwhile to include a few lines to state the purpose of feasibility study for those readers who may not understand this critical distinction.

- Why did you decide to include only participants with average pain intensity reported as moderate, severe, or very severe?

Response: We included patients with moderate pain intensity or more so that we can detect change on pain scores in the five point PROMIS pain intensity scale later in full clinical trial (i.e., to avoid floor effects). The two categories we excluded were "no pain" and "mild pain".

- Why did you choose to use the PROMIS five-point PROMIS Pain Intensity Short-form Scale?

Response: We recognize that many low back pain studies use numerical pain rating scales to assess pain intensity. However, in our previous study, Pathak et al., (2018, accepted for publication in Pain Reports pending minor revisions) found that many Nepalese adults with musculoskeletal pain failed to respond correctly to the numerical rating scales for pain, and much more Nepalese (especially with lower education levels and older individuals) preferred a verbal rating scale over a numerical rating scale. Similarly, in the validation study of numerical pain rating scale and our clinical practice, we found that many patients have difficulty responding to numerical scales. On the other hand, patients did not have difficulty in completing the PROMIS pain intensity short form scales in the validation study, which we think is more suitable for our context.

- In the description of the intervention in the education group, I believe you don't need to explain that education is supported by research and the available evidence in this section. So you should remove this paragraph.

Response: Thank you for this suggestion. We have removed the texts from the revised manuscript.

- I am not sure that you should call the intervention proposed as 'evidence based usual physiotherapy care'. Recent guidelines emphasize the importance of exercises. Why did you not consider the inclusion of exercise therapy in your treatment group? Once you call it as evidence based treatment, exercise has to be included. The same for manual therapy.

Response: Thank you for this important comment and question. We understand that exercise is

recommended as a primary treatment for low back pain by recent clinical practice guidelines. For this reason, the control group receives cycling as exercise during the one hour supervised treatment. Similarly, both the groups receive advice to walk up to 30 minutes (with rest periods if required), and advice to remain active and perform physical activities at home. No specific type of exercise is recommended by the clinical practice guidelines. Back-specific exercises and other forms of exercises will be provided to the patients after the reassessment at one week. Similarly, recent guidelines also recommend massage and manipulative therapy for low back pain. Therefore, based on the competence of the therapists working at the center, we delivered massage to the patients, although we excluded manipulative therapy.

Finally, we have renamed the control group treatment as “evidence-based physiotherapy treatment” (i.e., we removed the word “usual”) throughout the revised manuscript.

FORMATTING AMENDMENTS (if any)

Required amendments will be listed here; please include these changes in your revised version:

- Kindly re-upload SUPPLEMENTARY FILE in PDF format.

Response: We have uploaded the supplementary file in PDF format.

Additional changes:

1. We have revised the manuscript throughout for clarity and accuracy of English.
2. We have updated the references in Table 3, in the secondary outcome measures.

VERSION 2 – REVIEW

REVIEWER	Kory Zimney University of South Dakota, United States of America
REVIEW RETURNED	25-May-2018
GENERAL COMMENTS	My initial review points were adequately addressed in the revision.
REVIEWER	Matheus Almeida Universidade Cidade de Sao Paulo (UNICID), Brazil
REVIEW RETURNED	12-Jun-2018
GENERAL COMMENTS	In my opinion, the authors have satisfied most of my requirements. I am satisfied with their answers and amendments.